# Expression Pattern and Ligand Binding Characteristics Analysis of Chemosensory Protein SnitCSP2 from *Sirex nitobei*

**DOI:** 10.3390/insects14070583

**Published:** 2023-06-27

**Authors:** Pingping Guo, Enhua Hao, Han Li, Xi Yang, Pengfei Lu, Haili Qiao

**Affiliations:** 1The Key Laboratory for Silviculture and Conservation of the Ministry of Education, School of Forestry, Beijing Forestry University, Beijing 100083, China; gpingping0429@163.com (P.G.); hao452115308@126.com (E.H.); 18610268286@163.com (H.L.); yangxi1997000@163.com (X.Y.); 2Institute of Medicinal Plant Development, Chinese Academy of Medical Sciences and Peking Union Medical College, Beijing 100193, China

**Keywords:** chemosensory protein, *Sirex nitobei*, tissue expression, molecular docking, binding characteristics, dynamics simulation, molecular interaction

## Abstract

**Simple Summary:**

Since existing control strategies are not yet effective, in order to elucidate the molecular mechanisms of protein–ligand binding for effective pest control, three ligands that bind best to chemosensory protein 2 of *Sirex nitobei* were screened in this study. The host plant volatile (+)-α-pinene, symbiotic fungal volatiles terpene and (−)-globulol were found to bind most stably by molecular docking and dynamic simulations, and their free binding energies were calculated by the molecular mechanics Poisson–Boltzmann surface area method. Furthermore, some key amino acid residues were deeply explored, providing a favorable molecular basis for regulating the behavioral interactions of insects and developing new pest control strategies.

**Abstract:**

*Sirex nitobei* is an important wood-boring wasp to conifers native to Asia, causing considerable economic and ecological damage. However, the current control means cannot achieve better efficiency, and it is expected to clarify the molecular mechanism of protein–ligand binding for effective pest control. This study analyzed the expression pattern of CSP2 in *S. nitobei* (SnitCSP2) and its features of binding to the screened ligands using molecular docking and dynamic simulations. The results showed that SnitCSP2 was significantly expressed in female antennae. Molecular docking and dynamic simulations revealed that SnitCSP2 bound better to the host plant volatile (+)-α-pinene and symbiotic fungal volatiles terpene and (−)-globulol than other target ligands. By the molecular mechanics Poisson–Boltzmann surface area (MM-PBSA) method, the free binding energies of the three complexes were calculated as −44.813 ± 0.189 kJ/mol, −50.446 ± 0.396 kJ/mol, and −56.418 ± 0.368 kJ/mol, and the van der Waals energy was found to contribute significantly to the stability of the complexes. Some key amino acid residues were also identified: VAL13, GLY14, LYS61, MET65, and LYS68 were important for the stable binding of (+)-α-pinene by SnitCSP2, while for terpenes, ILE16, ALA25, TYR26, CYS29, GLU39, THR37, and GLY40 were vital for a stable binding system. We identified three potential ligands and analyzed the interaction patterns of the proteins with them to provide a favorable molecular basis for regulating insect behavioral interactions and developing new pest control strategies.

## 1. Introduction

Insects have a highly sophisticated olfactory system to distinguish various volatile chemicals from their prey, host plants, and conspecifics [1,2,3]. The discriminated olfactory stimuli are transmitted to the central nervous system via electrical signal transduction, producing a series of behavioral responses such as foraging, defense, courtship, reproduction, information exchange, and habitat selection to adapt to the surrounding environment [2,4]. Previous studies showed that the olfactory system of insects is hyper selective for semiochemicals, which are important in mediating behavioral patterns such as mate choice and food source choice [5,6]. Several classes of olfactory proteins were reported to be involved in chemosensory perception, including odor-binding proteins (OBPs), chemosensory proteins (CSPs), odor receptors (ORs), ionotropic receptors (IRs), and sensory neuron membrane proteins (SNMPs) [7]. In general, OBPs and CSPs represent two classes of functionally similar carrier proteins, usually located on antennae and mouthparts, which are major proteins involved in the recognition of volatiles and play an essential role in the transport of incoming odors to the corresponding receptors [8,9].

CSPs are the major binding proteins and are highly conserved evolutionarily and similar across species compared to OBPs [10]. The main difference with OBPs is that they bind and carry non-volatile odors and semiochemicals [11,12]. CSPs are all small spherical water-soluble proteins with a specific motif of four cysteines that form two disulfide bridges between neighboring residues, which are conserved [13,14]. The earliest identified members of the CSP family, isolated from the antennae of *Drosophila melanogaster*, were the olfactory-specific protein D (OS-D), the OS-D-like protein [15,16], and the pheromone-binding protein A-10 (A-10) [17]. Subsequently identified CSPs include CLP-1 [18], p10 [19], the sensory appendage protein (SAP) [16,20,21], and the CSPs themselves [22,23]. Since the first CSP family member was discovered, CSPs were identified for over a dozen insect species [24]. AgamCSP3 was abundant in the antennae of *Anopheles gambiae* and could bind to pheromones [25]. Highly expressed in antennae, *AlinCSP1-3* of *Adelphocoris lineolatus* showed relatively good binding affinity to Cis-3-Hexenol, Methyl salicylate, and n-Valeraldehyde [26]. *SinfCSP19* expressed in the male antennae of *Sesamia inferens* can successfully bind six host volatile components [27] and exhibit excellent binding capacity to three *S. inferens* pheromones, implying that SinfCSP19 was important in *S. inferens* host identification. These findings suggested that CSPs play essential roles in chemical signal transduction. As a result, it is critical to explore CSPs’ distinctive physiological roles.

*Sirex nitobei* is a wood-destroying wasp that interacts obligatorily with two wood-decaying fungi to disturb and weaken conifers. The species grew established in various Asian nations where it did not previously occur. *S. nitobei* was initially reported in China in 1980 and it since expanded to 13 provinces [28]. In 2016, the discovery of *S. nitobei* in Inner Mongolia was confirmed to result in the decline and mortality of a large amount of *Pinus sylvestris* var. *mongolica* [29]. *S. nitobei* has a relatively wide range of hosts, including *Pinus tabuliformis*, *P. sylvestris* var. *mongolica*, *Pinus massoniana*, *Pinus armandii*, and *Pinus thunbergia*. As a result, *S. nitobei* spread from its original discovery site in China to 1450 km northwest, 1750 km southwest, and 2200 km northeast [30]. As *S. nitobei* continues to expand its range, an increase in economic losses to forests will inevitably grow. To prevent host mortality and economic losses, it is important to elucidate the molecular interactions that occur in specific binding regions of SnitCSPs to develop and design efficient elicitors.

In previous studies, six *SnitCSPs* were identified using antennal transcriptome analysis [31]. In this study, CSP2 with relatively high fragments per kilobase million (FPKM) values obtained in the antennal transcriptome was selected. We first resolved its expression pattern in different tissues, and subsequent studies aimed to screen for active molecules that affect insect behavioral responses and explore the binding mode and properties of the SnitCSP2-ligand complex to provide new ideas for the development of novel attractants and pest management.

## 2. Materials and Methods

### 2.1. Sample Collection and Preservation

The affected wood segments for culturing S. nitobei in the experiment were obtained in Tongliao City, Inner Mongolia Province (China, 43°39′ N, 122°14′ E). One-meter-long injured wood segments with teardrop-like runoff points and premature aging were placed in a climate-controlled net cage at 25 ± 1 °C and 60–70% relative humidity. The wasps were caught soon after eclosion and time, sex, and status information were marked. We separated different tissues (the antennae, heads, thoraxes, legs, external genitals) quickly from the adults and placed them in RNA later buffer solution (Invitrogen, Carlsbad, CA, USA). Three sets of biological templates per tissue were collected and stored at 4 °C for 24 h and were then stored at −20 °C or −80 °C for long-term storage in the Forest Conservation Laboratory.

### 2.2. RNA Isolation and PCR Amplification

Total RNA was isolated from different tissues of both sexes separately (20 each from males and females) using Trizol reagent (Invitrogen, Carlsbad, CA, USA) following the manufacturer’s instructions. All the RNA samples were treated with RNeasy Plus Mini Kit (Qiagen, Hilden, Germany) to eliminate the genomic DNA. Using a NanoDrop ND-8000 spectrophotometer (NanoDrop products, Wilmington, DE, USA) to measure the concentration of isolated RNA, their integrity was determined by agarose gel electrophoresis. RNA quality was verified using a 2100 Bioanalyzer RNA Nanochip (Agilent, Santa Clara, CA, USA). The high-quality RNA samples (OD260/280 = 1.8–2.2, OD260/230 ≥ 2.0, RIN ≥ 6.5, 28S:18S ≥ 1.0, >10 μg) were placed at −80 °C and used to generate cDNA libraries. Synthesis of cDNA template was performed by PrimeScriptTM RT Reagent Kit with gDNA Eraser Kit (TaKaRa, Japan) for first-strand synthesis kit. PCR amplification was performed in a 20 µL volume containing 0.5 µL of each degenerate primer, 12.5 µL of Premix, and 10.5 µL of double distilled water (ddH_2_O). The cycling conditions were an initial denaturation at 95 °C for 2 min, followed by 34 cycles of 95 °C for 30 s, 55 °C for 30 s, and 72 °C for 1 min, followed by a final extension at 72 °C for 10 min and storage at 4 °C. Samples were sent to Rui Bo Xing Ke Biotechnology Company (Beijing, China) to complete sequencing.

### 2.3. Tissue Expression Pattern and Sequencing Analysis

The expression profile for different tissue of SnitCSP2 was performed by qRT-PCR. Gene-specific primers were designed by the software Beacon Designer 7.90 (PREMIER Biosoft International) to amplify the complete or partial ORF sequences of the CSP gene (Table 1). β-Tubulin was employed as a reference gene to normalize target gene expression and to correct for sample-to-sample variation. Each reaction was carried out in a 20 µL reaction mixture containing 0.5 µL of forward primer (5 µM), 0.5 µL of reverse primer (5 µM), 1.0 µL of sample cDNA (150 ng), 8 µL of nuclease-free H_2_O, and 10 µL of Mix (2×Taq PCR StarMix (TransGen Biotech, Beijing, China). The reaction schemes were as follows: 95 °C for 30 s, then 40 cycles of amplification (95 °C for 5 s and 60 °C for 30 s). The melting curve was analyzed for PCR products to detect a single gene-specific peak and to check for the absence of primer dimer peaks. Negative controls were non-template reactions (replacing cDNA with H_2_O). All experiments had three technical replicates and three biological duplicates. The comparative 2^−ΔΔCt^ method was applied to calculate the relative quantification of different tissues [32]. Comparative analyses of target gene among different tissues were determined using one-way ANOVA tests followed by Tukey’s HSD method using SPSS statistical software (version 26.0, SPSS Inc., Chicago, IL, USA) (*p* < 0.05). When applicable, the values are shown as the mean ± SE. The SnitCSP2 sequence (GenBank: QHN69081.1) was accessed based on the transcriptome sequencing data of the S. nitobei antennae and subsequent bioinformatic analyses were conducted to enhance the knowledge of SnitCSP2. Sequence homologous alignment and similarity searches were carried out by Blast biological software (http://www.ncbi.nlm.nih.gov/blast (accessed on 6 October 2022)). The presence of a signal peptide was predicted using the artificial neural network algorithm of SignalP5.0 (http://www.cbs.dtu.dk/services/SignalP-5.0/ (accessed on 6 November 2022)). The physical and chemical properties of SnitCSP2 were predicted using the online program Expasy.

### 2.4. Homology Modeling and Molecular Docking

The 3D model of SnitCSP2 was calculated with SWISS-MODEL (https://www.swissmodel.expasy.org/ (accessed on 26 November 2022)) [33] and SAVES 6.0 (https://saves.mbi.ucla.edu/ (accessed on 26 November 2022)) was utilized to evaluate the quality of the constructed models [34,35,36]. In this study, the CSP6 crystal structure in *Mamestra brassicae* (PDB ID:1KX9, chain A) was used as the best modeling template to construct a reasonable protein model. We screened 14 odor ligands including sex pheromones, host volatiles, and symbiotic fungal volatiles (Table 2), whose 3D structures (Figure 1) were searched via access to the NCBI PubChem (https://pubchem.ncbi.nlm.nih.gov/ (accessed on 28 November 2022)) website by name or CAS number. Autodock 4.2.6 was then used to practice the semi-flexible docking between SnitCSP2 and 14 ligands. Each protein–ligand complex binding pose was scored by the score–ligand pose tool based on their plausibility, free binding energy, and other factors. Eventually, the complex with the best conformation that had lowest free binding energy was chosen. PyMOL 2.2.0 software (Schrodinger, San Diego, CA, USA) was used to produce and further analyze images.

### 2.5. Molecular Dynamics Simulation of the SnitCSP2-Ligand Complexes

After obtaining the protein models and small molecule ligands, production runs were performed using the Gromacs 2019.6 package [37] via the ACPYPE script [38], using the amber99sb-ildn force field [39] for the SnitCSP2 and the AmberTools force field (GAFF) 18 [40] for ligands. In each simulation, the protein was placed in a periodic cubic box, and the minimum distance between the protein and the box was set to 1.0 nm. Na^+^ and Cl^−^ were incorporated to neutralize the system during the dissolution of SnitCSP2-ligand complexes by water molecules with a three-point model of transferable interatomic potential (TIP3P). This simulation used the steepest descent algorithm for energy minimization of the system. The V-rescale thermostat was used for the temperature coupling, and the Parrinello–Rahman barostat was used for the pressure coupling. Canonical (NVT) and isothermal-isobaric (NPT) ensembles were used to equilibrate the systems. Then, 40 ns molecular dynamics (MD) simulations with a time step of 2 × 10^−3^ ps were performed to analyze the conformational translocation of the complex. Simulation experiments were repeated in an identical environment to check reproducibility. QTGRACE for analysis of dynamics results such as the root mean square deviation (RMSD) and the root mean square fluctuation (RMSF), PyMOL 2.2.0 for visualization of complex structures. Bimolecular interactions were calculated by gmx-PBSA based on the molecular mechanics Poisson–Boltzmann surface area.

## 3. Results

### 3.1. Identification and Analysis of SnitCSP2

The results indicated that the open reading frame (ORF) of SnitCSP2 was 375 bp long and encoded 124 amino acids (Figure 2), and its amino acid sequence contains four conserved cysteine sites. The various characteristics are listed in Table 3. Lysine (Lys) accounted for the highest percentage of the protein’s 20 amino acids, accounting for 11.3%, and the instability index (>40) revealed that SnitCSP2 may be unstable. The highest sequence similarity between SnitCSP2 and SnocCSP2 was 99.19%, and there were dozens of CSPs with sequence similarity above 60% after checking the NCBI database, indicating that there was a high consistency between the sequences of these insects.

### 3.2. Tissues Expression Pattern Analysis

The highest expression of SnitCSP2 was in female antennae, followed by female legs, and lowest in male legs, with antennae expressing much more than other tissues (Figure 3). SnitCSP2 expression was 3.05 times higher in the antennae than in the legs. The previous studies concluded that chemoreceptive proteins are mostly expressed in the antennae, implying a function in odorant recognition within antenna and transmission of chemical stimulus signals, suggesting that SnitCSP2 is likely to be involved in the olfactory behavior of the *S. nitobei*. Furthermore, SnitCSP2 expression was 1.39-fold higher in female antennae than in male antennae, showing a sex bias that may point to differences in its roles in males and females, such as searching recognition of the opposite sex or females searching for egg-laying hosts.

### 3.3. Modeling and Model Evaluation

The CSP6 crystal structure in *M. brassicae* (PDB ID:1KX9, chain A) had 42.45% adequate homology (sequence identity > 30.0%) with SnitCSP2 (Figure 4B), and after the screening, we chose it as the best modeling template to construct a reasonable protein model. To ensure the accuracy of the subsequent experiments, we evaluated the stereochemical quality of the constructed model (Figure 4A) to determine its reliability. In the Ramachandran plot of the SnitCSP2 model (Appendix A), 92.6% (>90%) of the amino-acid residues fell in the most favored regions, and in addition, the total quality factor of ERRAT was 98.958, while 100% of the residues met the Verify_3D criteria (Appendix A), which exactly indicated that the model had a great quality to perform molecular docking.

### 3.4. Binding Affinities for SnitCSP2 with Ligands and Molecular Docking

The binding energy reflects the stability of binding, and a small binding energy indicates a more stable binding. For both pheromones and host volatiles, SnitCSP2 exhibited a greater focus and a narrower binding energy fluctuation under the identical source ligands (Figure 5). SnitCSP2 showed the lowest affinity for binding to sex pheromones. However, the binding energy for symbiotic fungus volatiles differed greatly, with 2-hexene having the greatest binding energy of −3.34 kJ/mol and (−)-globulol having the lowest binding energy of −6.46 kJ/mol, suggesting that SnitCSP2 has a larger affinity for particular specific fungal volatiles. Furthermore, three ligands with binding energies lower than −5.0 kJ/mol were screened, namely, host plant volatiles (+)-α-pinene, symbiotic fungal volatiles terpene and (−)-globulol, with binding energies of −5.04 kJ/mol, −5.85 kJ/mol, and −6.46 kJ/mol, respectively, showing that they bind more stably.

### 3.5. Stability of SnitCSP2-Ligand Complexes in MD Simulation

Utilizing the MD trajectories generated, the root mean square deviation (RMSD) and the root mean square fluctuation (RMSF) were computed to analyze the stability of the docked complexes, and the results are discussed below. The RMSD curves for all complex systems showed that the 40 ns MD simulations have implications that the overall structure can be explored, as they all converged within 40 ns. Almost all of the docked complexes reached an equilibrium around 20 ns with varying average RMSD. The maximum and minimum values of RMSD standard deviation for these 14 systems were 0.07 nm and 0.01 nm, respectively, and the average RMSD values varied from 0.26 nm to 0.52 nm (Figure 6). Apart from allowing to the assessment of the equilibration, quality of the run, and convergence of MD trajectories, RMSD is useful to investigate the stability of a protein in a complex. A larger RMSD value is indicative of the lower stability of a complex. The average RMSD value for the SnitCSP2-terpene complex was approximately 0.26 nm, and during the 40 ns simulation period, almost no significant fluctuations were seen, demonstrating enhanced complex stability across the entire dynamics. Average RMSD values for CP-1, CP-2, CP-3, CF-3, and CF-4 were higher than 0.40 nm, and several of them were even near 0.5 nm with more pronounced oscillations. As a result, they were demonstrated to be among the least stable complexes, which is consistent with the findings of molecular docking.

The RMSF, which represents the features and degrees of freedom in the local movement of protein secondary construction elements when paired with the ligands, was used to further identify the local motion features of amino acid residues when SnitCSP2 was complexed with ligands. During the 40 ns MD simulations, the 14 complexes exhibited similar movements. Among them, the maximum RMSF value (1.09 nm) was discovered on ILE107 with a high degree of freedom in the CF-4 complex. Overall, there were three major regions with general dramatic fluctuations in the RMSF image, including residues 4–11 (N-terminus), 21, and 59–68 (loop at the front of the α4 and α4) and 104–107 (C-terminus), regions that are very near the SnitCSP2-ligands binding site and where the complex interactions are not stable. The dramatic variations were caused by the low stability of complex interactions. Nonetheless, there were moderately stable areas at residues 23–27, which were particularly near to the binding site in the α2 helix. Their existence was thought to be one of the most important aspects in sustaining the binding’s stability. In addition, CF-2 had modest peaks at significant fluctuation areas, with an average value of about 0.17 nm for the RMSF, which is consistent with the RMSD study and suggests that the complex is relatively stable (Figure 7).

### 3.6. Energy Calculation and Binding Modes Analysis

Based on the above findings, we selected the three most stably bound systems (C_H_-1, C_F_-2, and C_F_-7) for further investigation. The free binding energies of the relatively stable orbital data intercepted during the 40 ns molecular dynamics simulations were calculated using the MM-PBSA method and subdivided into four components for specific analyses: van der Waals energy (G_vdw_), electrostatic energy (G_ele_), polar solvation energy (G_PB_), and nonpolar solvation energy (G_SA_) (Table 4). Van der Waals and nonpolar solvation (SASA) energies were the key contributors to the C_H_-1 and C_F_-7 complexes, with electrostatic energy also preferring this combination, but their contribution was minor. Van der Waals energy, electrostatic energy, and SASA energy were the primary guarantees for system stability in C_F_-2. Van der Waals forces were the most important of the three driving forces in the binding of SnitCSP2 to the ligand. The presence of polar solvation energy, on the other hand, was harmful to protein–ligand binding, being the dominant barrier to protein–ligand binding, particularly for SnitCSP2 binding to C_F_-2, as demonstrated by the predicted positive value of G_PB_.

After calculating the binding energy of the whole complex, the decomposition of the energy contribution of each residue to the total binding energy was estimated (Figure 8) using the gmx-MMPBSA tool and the Python script (MmPbSaDecomp.py). The specific data were also visualized, and the residues with binding energies below −2.0 kcal/mol are annotated separately in Figure 9, including VAL13, GLY14, LYS61, MET65, LYS68, ILE16, ALA25, TYR26, CYS29, THR37, GLU39, GLY40 TYR98, LEU101, where the highest value of ∆G_bind_ was −7.662 kcal/mol and the lowest value was −2.201 kcal/mol. It was discovered that the RMSF values of the above binding residues were frequently low in the 40 ns MD simulations by monitoring the local motion features of amino acid residues. The RMSF values of GLY14, TYR26, CYS29, GLY40, TYR98, and THR37 were all less than 1.0 nm, while the RMSF value of ALA25 was much lower, at 0.06 nm, and they were all positioned in the RMSF curve’s trough area, which was positive for stable binding ligand. During binding, the area including LYS61 and MET65 experienced significant changes, perhaps leading to particularly unstable connections between them and the ligand. Other locations, such as the N-terminus, C-terminus, and a small peak around ASN59, were higher and unstable. In conclusion, these key amino acid residues can interact with small ligand molecules to form stable complexes.

The specific states and interactions of the three highly active ligands were highlighted by taking into account the findings of the related energy calculations as well as the precise locations in the protein (Figure 10). As shown, VAL13, GLY14, LYS61, MET65, and LYS68 of SnitCSP2 were shown to be strongly linked to (+)-α-pinene. Terpenes were firmly bound by ALA25, TYR26, CYS29, ILE16, THR37, GLU39, and GLY40, whereas (−)-globulol was significantly bound by TYR98 and LEU101. Among them, THR37 and TYR26 are polar amino acids, while the remainder are non-polar amino acids. The ligands in the binding cavity are closely controlled by these hydrophobic residues. Meanwhile, these amino acid residues interacted with ligands in a variety of ways (Figure 11), the most prevalent of which were van der Waals and π–alkyl. It is worth noting that THR37 and the hydroxyl group of the terpene established a typical hydrogen bond, which aids in the stability of the C_F_-2 complex system.

## 4. Discussion

Chemosensory proteins are essential in the physiological responses of insects to identify external odor molecules. Insect CSPs are evolutionarily conservative, with sequence similarity reaching more than 40% across species. This trait may be linked to their binding properties, which are lower for common environmental volatiles and stronger for communication-related signaling chemicals. In this study, we observed that SnitCSP2 consists of six tapered arrangements of α-helices (α1-α6) by constructing a three-dimensional model of it. It is considered to bind a greater spectrum of ligand molecules due to its varied three-dimensional structure and a vast range of sequence lengths. Two disulfide connections created between four conserved cysteines (Cys29- Cys36, Cys55- Cys58) connect α2 and α3, α3 and α4, in pairs, which can expand the binding pocket to bind larger proteins and stabilize the correct conformation of SnitCSP2. These qualities also provide a molecular foundation for further research into the use of CSP as pest management targets.

In insect olfactory receptors, chemosensory proteins bind hydrophobic odor molecules, transport them across the hydrophilic lymph fluid, and convey them to olfactory neurons, converting chemical impulses into electrical signals. *CSPs* are abundantly expressed in the olfactory receptors of insects such as *Polistes dominulus* and *Linepithema humile,* most of which have high expression levels mainly in the antennae, implying that CSPs play an essential function in the olfactory system [41]. According to Li et al.’s study on *Pieris rapae*, *PrapCSP16* is primarily expressed in female antennae which may perceive chemical information involved in host localization [42]. *CcunCSP1* and *CcunCSP3* of *Chouioia cunea,* as well as *MmedCSP2* and *MmedCSP3* of *Microplitis mediator*, were largely expressed in the antennae, and *CcunCSP3* was obviously more expressed in female antennae than in male antennae, showing a sex difference in expression levels [43,44,45]. In this study, *SnitCSP2* was found to be strongly expressed in female antennae, presumably involved in chemoreception, which may be associated with behaviors like locating target spawning sites and sensing other chemical stimuli. Moreover, *SnitCSP2* was found to be substantially more expressed in female antennae than in male antennae, indicating a gender-specific difference in the expression level, which is consistent with the tissue expression profile of *CcunCSP3*. It is worth noting that an in-depth study into the function of CSP highly expressed in antennae could help explain the mechanism of interaction between *S. nitobei* and the external environment as well as within the population, and potentially acts as a target gene to interfere with insect olfactory recognition behavior.

CSPs have a wide range of expressed tissues and are commonly expressed not only in olfactory organs but also in non-olfactory organs, with broader functional and investigational significance, such as the abdomen [46], reproductive organs [47], and feet [48]. *Sesamia inferens* transcriptome sequencing revealed 24 CSP genes, seven of which (CSP2, CSP5, CSP6, CSP7, CSP16, CSP20, and CSP23) were highly expressed in larvae and gonads [49]. Li et al. found that PrapCSP20 was enriched in the testes of *Pieris rapae* and probably participated in the reproduction of insects [42]. A variety of CSPs were found in insects such as *Hylamorpha elegans* (HeleCSP3) [50], *Ophraella ommuna* Lesage (OcomCSP12) [51], and *Scopula subpunctaria* Herrich-Schaeffer (SubCSP1/16) [52] that may be involved in insect egg-laying behavior. Therefore, we cannot exclude that SnitCSP2 plays other roles in *S. nitobei*. CSPs have distinct functions in different developmental stages, sexes, and tissues. Notably, the same CSP may express itself simultaneously in several organs or tissues, indicating that it may perform multiple functions. In the current study, we found that *SnitCSP2* was also expressed in the head, thorax, external genitalia, and legs, in addition to the antennae. Such a wide expression demonstrates that it is crucial in the biological behaviors of insects, implying that SnitCSP2 may have multi-functions and affect various physiological behaviors of *S. nitobei*. Moreover, it was found that *SnitCSP2* was relatively highly expressed in female legs, suggesting that it has a specific purpose for the legs.

Depending on the expression level of CSPs in different tissues, the selection of environmental odor molecules is specific. Correspondingly, organisms influence the behavioral response of biological olfaction by modulating the presence or concentration of certain odor molecules from a molecular perspective, which, in turn, manifests itself in behavioral interactions. For instance, *SinfCSP19* of *Sesamia inferens* possessed abundant expression in male antennae and exhibited excellent binding ability to three sex pheromone components but also to six host volatile components [27], indicating that SinfCSP19 was engaged in *S. inferens*’ host recognition. Significantly expressed in the antennae of *Adelphocoris lineolatus*, AlinCSP1-3, was capable of binding chemical ligands such as n-Valeraldehyde, Methyl Salicylate, and Cis-3-Hexenol [26]. Foret et al. [53], in their study on *Apis mellifera*, found that AmelCSP5 had a critical function during fertilized egg growth. The gene encoding this protein was discovered solely in the queen’s eggs and ovaries, and was not observed in the body parts of any other larvae or adult. When this gene was knocked out, fertilized eggs do not develop fully and the eggs do not hatch [54].

Compared to conventional chemical ecology, the reverse chemical ecology strategy provides a rapid and low-cost way to screen odor molecules with potential attraction or tropism activity. The development of this method is based on understanding the molecular processes of the olfactory system of insects and the ability of odorant proteins to bind to behaviorally active chemicals [55]. According to the local energy search and Lamarckian genetic algorithm, the lower binding energy indicates a better binding affinity of the protein to the ligand [56]. Chen et al. investigated the binding of SfurCSP5 from *Sogatella furcifera* with host rice volatiles and discovered that SfurCSP5 had a strong affinity with 3-tridecanone, 2-pentadecanone, and β-lonone, and similarly, SfurCSP5 had the lowest molecular docking scores with the three substances [57]. Tian et al. used three-dimensional modeling and molecular docking to predict the affinity of the sex pheromone binding protein CpomPBP2 from *Cydia pomonella* with 35 odor molecules and discovered the strongest binding capacity for 1-dodecanol [58]. Venthur et al. verified the binding properties of the sex pheromone binding protein LbotPBP1 from *Lobesia botrana* to six host volatiles and eleven *L. botrana* pheromones using molecular docking and dynamics simulations, indicating that 1-dodecene was the best ligand for Lbot PBP1 [59]. These findings will help with a future investigation into the mechanism of interaction between SnitCSP2 and odorant chemicals.

In this study, molecular docking of the SnitCSP2 protein with 14 odor ligands revealed that three small molecule chemicals, (+)-α-pinene, terpene, and (−)-globulol, showed low binding energy. Previous studies demonstrated that volatiles from host trees attract tree wasps with weaker hosts are more attractive [60]. *Pinus sylvestris* var. *mongolica* is the primary host plant of *S. nitobei* [61], which emits the volatile substance (+)-α-pinene [62]. In forest behavior experiments, plant-sourced trap cores with a mixture of (+)-α-pinene, 3-carene, and camphene successfully captured numerous *Sirex. noctilio* [63], which is a species closely related to *S. nitobei*. In China, *S. noctilio* appeared in the field from the end of June to the beginning of September, after which the same trees were found to produce peaks of *S. nitobei* in late August and late September, during a period when they would co-infest the host plant species. In this research, SnitCSP2 exhibited a strong binding affinity for these ligands, further revealing its function in host plant recognition.

*A. areolatum* and *A. chailletii* are symbiotic fungi of *S. nitobei*. The female woodwasps deposit a phytotoxic mucus and an obligate symbiotic fungus, *Amylostereum areolatum* (Fr.) Boidin (Basidiomycotina: Corticiaceae), in the trees at the time of oviposition, causing damage [64,65]. The toxic mucus encourages the formation of symbiotic fungal spores, allowing the fungus to occupy the new niche in host trees quickly [64]. The growth of the symbiotic fungus is correlated with the development of woodwasp larvae who feed exclusively on the fungus until the third instar, and then on fungus-colonized wood [66]. As a result, insects, toxins, and fungi all collaborate to harm host trees. Adult woodwasps do not feed, instead, they utilize nutrients stored during the larval stage. Therefore, female woodwasps must find a suitable host before laying eggs. They probe the sapwood with their ovipositor for a favorable growth environment for the development of their young and the symbiotic fungus. In response to the behavior of *S. nitobei* in discovering optimal oviposition sites via symbiotic fungi, the integration of chemical trap cores of plant sources with symbiotic fungal volatiles can effectively trap female wasps attempting to lay eggs. Terpene and (−)-globulol, two symbiotic fungus volatiles, exhibited a strong affinity with SnitCSP2. Both volatiles were highly appealing to female tree wasps [67], and as close relatives, male *S. nitobei* probably favors regions with high coprophilous fungal volatile concentrations due to the presence of more debilitated host plants and more abundant mating resources in these areas. Molecular docking and dynamic simulation research demonstrated that terpene and (−)-globulol can bind firmly to SnitCSP2, which may be connected to adult male mating behavior. This protein attaches to symbiotic fungus volatiles and signals to other males the possible presence of females. Consequently, by manipulating the amount of terpene and (−)-globulol, it may be feasible to manage behavioral interactions between males and their surroundings.

Analyzing the interaction forces of SnitCSP2 with three small molecule chemicals and predicting the key amino acid binding sites can further probe the binding mechanism of SnitCSP2 to odor molecules. In this research, we found that the force types of SnitCSP2 amino acid residues were mainly van der Waals forces, but the polar solvation energy plays an obstructive role in binding. Importantly, the binding of diverse ligands differs significantly, probably due to particular amino acids within the binding cavity. For example, for the binding of CSPsg4 to the target ligand, IIE76, and TRP83 were critical for the binding of oleamide [68], and TYR26 was essential for 12-bromo-dodecanol (BrC_12_OH) binding in CSPMbraA6 [69]. Thus, VAL13, GLY14, LYS61, MET65 and LYS68 were important for the stable binding of (+)-α-pinene by SnitCSP2, ILE16, ALA25, TYR26, CYS29, GLU39, THR37, and GLY40 were vital for a stable binding system for terpenes.

Based on reverse chemical ecology, CSPs are involved in insect olfactory behavior by binding to odor molecules, suggesting that it could serve as an entry point for developing innovative ecologies, providing a basis for developing tools to interfere with or modulate biological interactions. In this study, the expression characteristics of the SnitCSP2 gene from *S. nitobei* were determined, and three potential odor molecules and key amino acid binding sites were identified and screened using computer simulations, providing new ideas to explain the molecular mechanism of protein–ligand binding and subsequent pest management, with the possibility of novel attractants developed by chemical ecology principles. It also offers a theoretical foundation for the subsequent validation of Snit CSP2 in vivo function by fluorescence competition binding assays and gene editing techniques, and, thus, for the design of pest control strategies based on the regulation of *S. nitobei* chemical communication.

## Figures and Tables

**Figure 1 insects-14-00583-f001:**
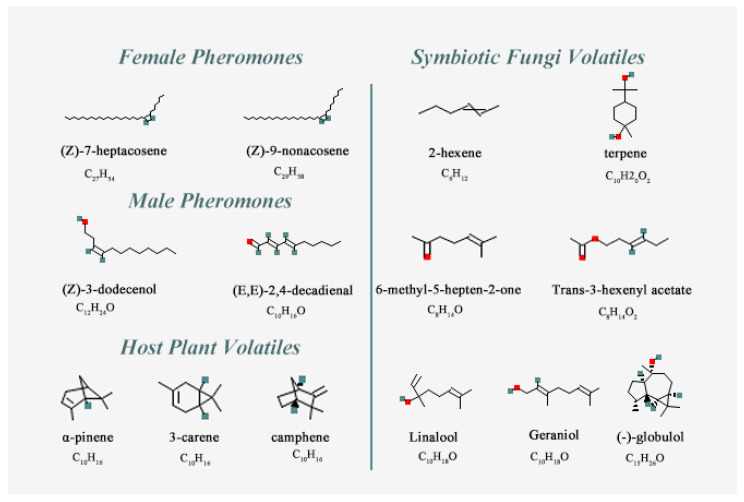
The two-dimensional structure information of the ligands.

**Figure 2 insects-14-00583-f002:**
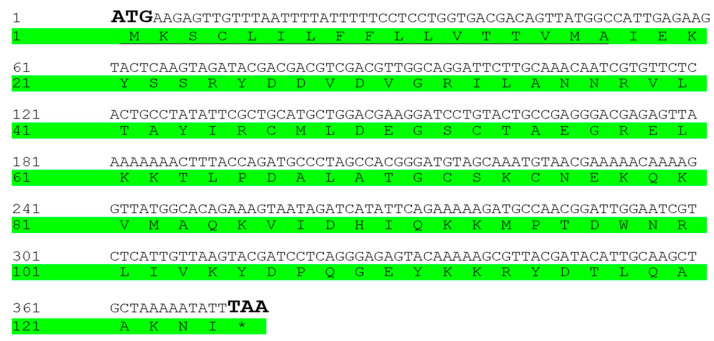
Nucleotide and deduced amino sequences. The start and stop codons are shown in bold type, the asterisk marks the translation termination codon, and the predicted signal peptide is underlined.

**Figure 3 insects-14-00583-f003:**
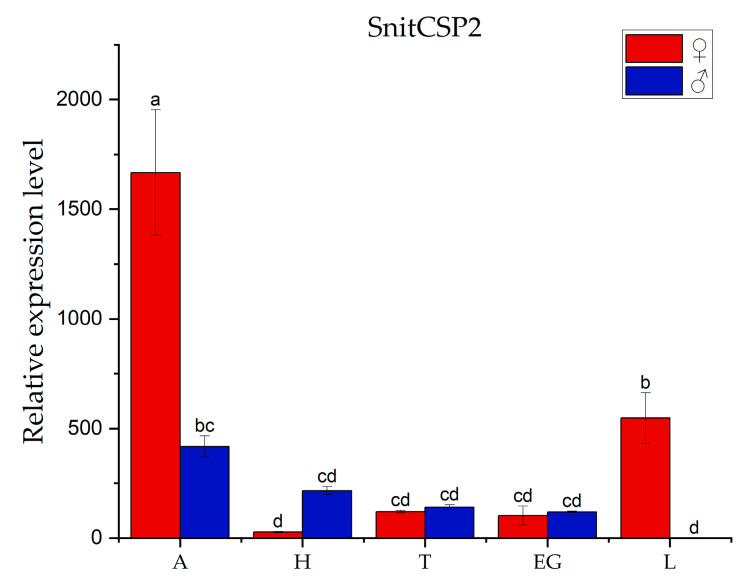
Transcript levels of tissue-specific CSP gene in different tissues of *S. nitobei*. A, antennae; H, head (without antennae); T, thorax; EG, external genitalia; L, leg. Red indicates expression in females and blue indicates expression in males. The reference gene β-tubulin was used to normalize CSP gene expression and correct for sample-to-sample variation. Transcript levels were normalized to those of FT. The error bar represents the standard error, and the lower cases above each bar indicate significant differences (*p* < 0.05).

**Figure 4 insects-14-00583-f004:**
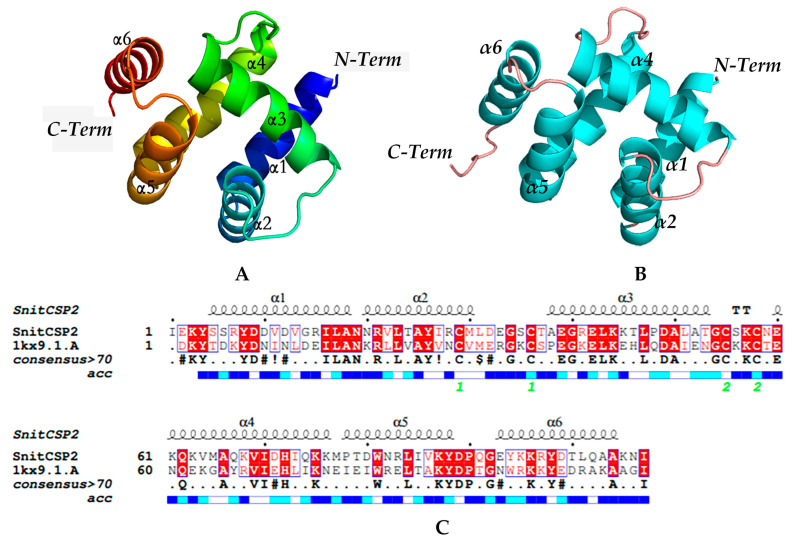
(**A**) Model of SnitCSP2. (**B**) Corresponding modeling template of SnitCSP2. (**C**) Sequence comparison of SnitCSP2 with its modeling template.

**Figure 5 insects-14-00583-f005:**
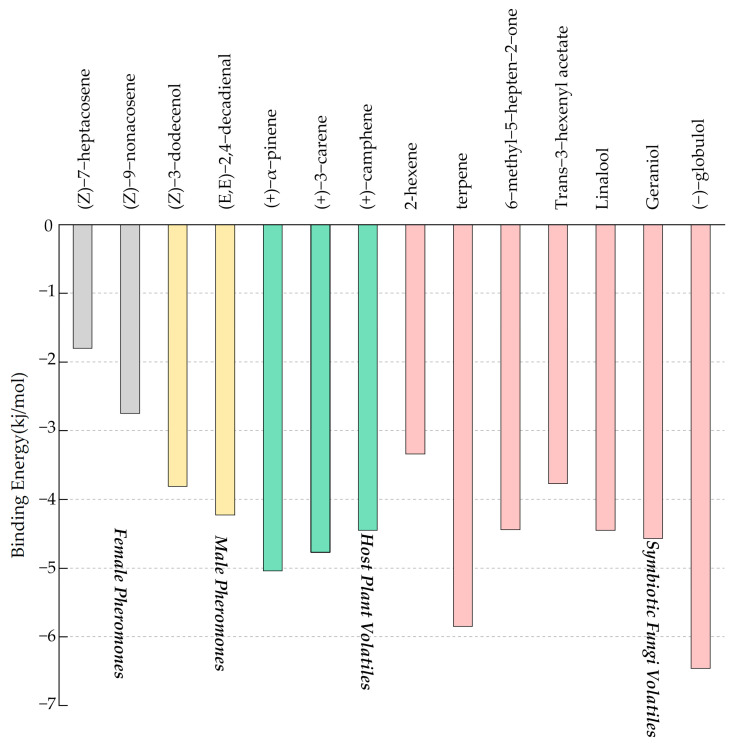
Binding energy distribution of SnitCSP2 with ligands docked.

**Figure 6 insects-14-00583-f006:**
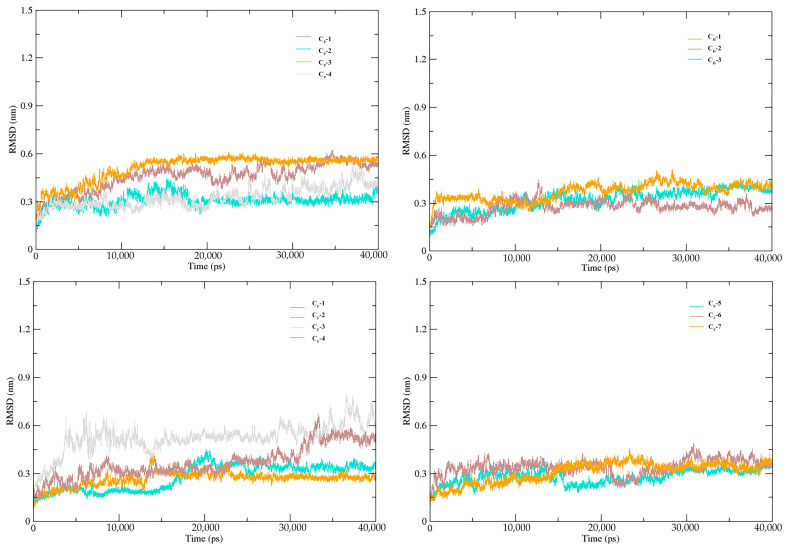
Time-evolving RMSD curves (40 ns) of SnitCSP2-ligand complexes.

**Figure 7 insects-14-00583-f007:**
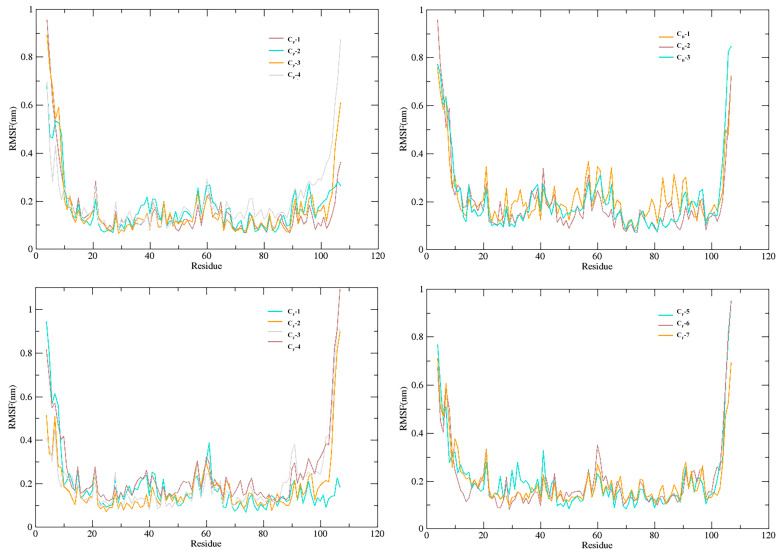
The root mean square fluctuation (RMSF) curve in MD simulations (40 ns).

**Figure 8 insects-14-00583-f008:**
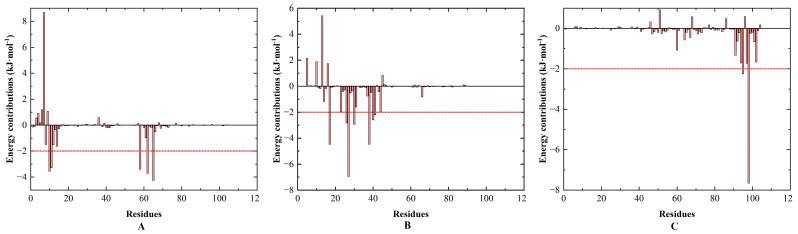
Residue energy contribution decomposition of SnitCSP2-ligand complexes. (**A**) C_H_-1, (**B**) C_F_-2, (**C**) C_F_-7. The vertical and horizontal axes represent the energy contribution and the energy of the corresponding residue, respectively.

**Figure 9 insects-14-00583-f009:**
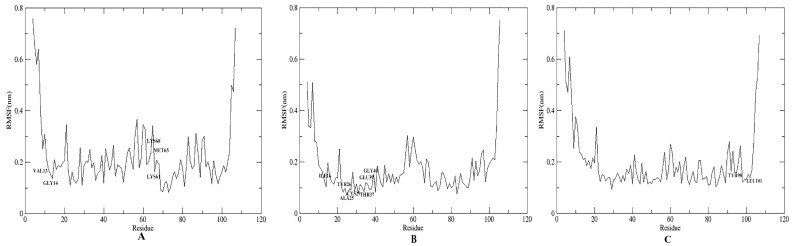
The RMSF curve of SnitCSP2-ligand complexes. (**A**) C_H_-1, (**B**) C_F_-2, (**C**) C_F_-7.

**Figure 10 insects-14-00583-f010:**
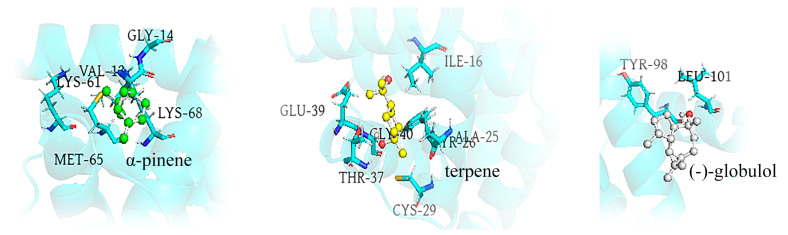
Canonical conformations of SnitCSP2-ligand complexes. Representative hydrophobic residues on the binding interface are marked. The ligands are displayed in the form of a sphere-stick model.

**Figure 11 insects-14-00583-f011:**
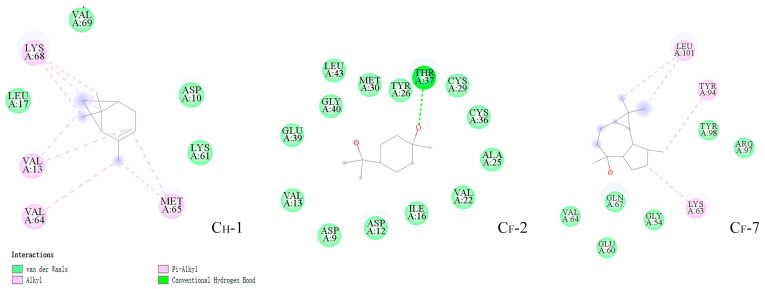
Interactions in C_H_-1, C_F_-2, and C_F_-7 complexes.

**Table 1 insects-14-00583-t001:** Primer information used in qPCR.

Gene Name	Primers Sequence (5′→3′)	Annealing Temperature, °C	Segment Length, bp
SnitCSP2	F: GTGACGACAGTTATGGCCATTG R: TGCAGCGAATATAGGCAGTG	59.0	106
β-Tubulin	F: CGTCGGTTCCGTTGATAAGTTG R: AGAATATCCCGACCGAGTGTTG	59.0	122

**Table 2 insects-14-00583-t002:** Information of 14 ligands and corresponding complexes.

Chemical Name	PubChem CID No.	Molecular Formula	Complex Code	Chemical Name	PubChem CID No.	Molecular Formula	Complex Code
** *Female Pheromones* **				** *Male* ** ** *Pheromones* **			
(Z)-7-heptacosene	56936088	C_27_H_54_	C_P_-1	(Z)-3-dodecenol	5364626	C_12_H_24_O	C_P_-3
(Z)-9-nonacosene	14367299	C_29_H_58_	C_P_-2	(E,E)-2,4-decadienal	5283349	C_10_H_16_O	C_P_-4
** *Host Plant Volatiles* **				** *Symbiotic* ** ** *Fungi Volatiles* **			
α-pinene	6654	C_10_H_16_	C_H_-1	2-hexene	19966	C_6_H_12_	C_F_-1
3-carene	26049	C_10_H_16_	C_H_-2	terpene	6651	C_10_H_20_O_2_	C_F_-2
camphene	92221	C_10_H_16_	C_H_-3	6-methyl-5-hepten-2-one	9862	C_8_H_14_O	C_F_-3
				Trans-3-hexenyl acetate	5352557	C_8_H_14_O_2_	C_F_-4
				Linalool	6549	C_10_H_18_O	C_F_-5
				Geraniol	637566	C_10_H_18_O	C_F_-6
				(−)-globulol	12304985	C_15_H_26_O	C_F_-7

C_P_, C_H_, and C_F_ refer to the complexes of SnitCSP2 and sex pheromones, host plant volatiles, and symbiotic fungi volatiles, respectively.

**Table 3 insects-14-00583-t003:** Information on the biochemical properties of SnitCSP2.

Name	Molecular Formula	MW (ku)	pI	Arg + Lys	Asp + Glu	Aliphatic Index	Instability Index	GRAVY
SnitCSP2	C_625_H_1024_N_172_O_184_S_10_	14.21	9.10	21	15	88.87	54.11	−0.346

**Table 4 insects-14-00583-t004:** Binding energy components of the complexes.

The Binding Energy Components	C_H_-1 (kJ/mol)	C_F_-2 (kJ/mol)	C_F_-7 (kJ/mol)
ΔE _VDW_	−53.189 ± 0.154	−99.712 ± 0.363	−71.014 ± 0.342
ΔE _elec_	−0.539 ± 0.002	−9.687 ± 0.036	−0.581 ± 0.003
ΔG _PB_	19.537 ± 0.056	68.209 ± 0.247	24.826 ± 0.119
ΔG _SA_	−11.559 ± 0.034	−16.699 ± 0.061	−12.959 ± 0.062
T ΔS _S_	−0.937 ± 0.003	−7.443 ± 0.027	−3.310 ± 0.016
ΔG _bind_	−44.813 ± 0.189	−50.446 ± 0.396	−56.418 ± 0.368

## Data Availability

The authors confirm that the data supporting the findings of this study are available within the article and its Appendix A.

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
