# Peer review of "Expression Pattern and Ligand Binding Characteristics Analysis of Chemosensory Protein SnitCSP2 from Sirex nitobei"

_insects, 2023, doi:10.3390/insects14070583_

Round 1

Reviewer 1 Report

This is a fairly high-quality manuscript that describes the expression pattern and ligand binding characteristics of a chemosensory protein (which the authors refer to as SnitCSP2) from Sirex nitobei, an insect pest of conifers native to Asia. It was discovered that SnitCSP2 was significantly expressed in female antennae; also, docking study results and dynamic simulations confirmed that SnitCSP2 bound well to the host plant volatile (+)-alpha-pinene and symbiotic fungal volatiles terpene and (-)-globulol. The supporting information is helpful and illustrative.

I detected no major methodological problems with the work described in the manuscript, and after several editorial issues are remedied (see below), I believe it is publishable.

-Title (lines 2 and 3) I would suggest “Ligand” not “Ligands” also there is no need to put a period after Sirex. It should be Sirex nitobei or S. nitobei, not Sirex. nitobei. This occurs elsewhere in the manuscript as well.

-line 22, “SnitCSP2 bound better to the host plant volatile…” Better than what? This sentence seems incomplete.

-line 36, “semiochemicals” not “semi-chemicals” unless I misunderstand. Again on line 47.

-line 37, suggest “food source choice.” not “food sources.”

-line 48, “soluble” …in what? Water? This should be specified.

-line 49, “which are more conserved.” More conserved than what?

-line 59, omit “the”

-line 70 and elsewhere, remove the period after Pinus.

-line 84, suggest “intend” instead of “intended”.

-line103, “All” not “all”.

-line 104, insert a space after “Kit”.

-line 113, ddH2O presumably means distilled, deionized water. This could be specified.

-line 127, omit “the” from “95 °C for the 30s”.

-line 157, what is meant by “the rational conformation”? This could be explained in a little more detail.

-line 214, Figure 4. (A) “Model of SnitCSP2” would suffice, not “Modeling model of SnitCSP2.”

-line 231, more closely than what? This could be explained it a little more detail.

-line 237, insert “the” before “terpene”.

-line 269 and 270, insert a space after “deviation” and “fluctuation”.

-line 272, “had the sense” doesn’t sound right. It is unclear what the authors mean with this sentence.

-line 297, insert a space after 107.

-line 311, suggest “reaching” not “can reach”.

-line 360, CSPs not CSPS.

-line 365, omit “the”

-line 368, suggest “such” instead of “like”.

-line 388, “pheromones” not “pheromone”.

-line 394, “…low binding energy and high binding energy…” seems unclear. Was the binding energy high or low?

-line 396, I would suggest rewording “…weaker hosts performing better attraction.” to “…weaker hosts are more attractive.”

-line 423 suggest “dynamic” not “dynamics”.

There are several editorial comments and typos which I have noted in my comments to the authors, but once these are corrected, the quality of the English language should be fine.

Author Response

Dear Reviewer:

        Thank you very much for your kindly comments on our manuscript. There is no doubt that these comments are valuable and very helpful for revising and improving our manuscript. The editorial comments and typos in the article have been corrected, thank you again for your valuable comments.

Reviewer 2 Report

In this article the authors have analyzed the expression pattern of SnitCSP2 and concluded that it is highly expressed within antenna. They also have executed computational screening of the ligands. However, the quality of computational protocol is questionable due to following reasons:

Simulation protocol is not correct. Production is done straight after the model building which is normally executed after minimization, heating and equilibrium.

The docking scores are not promising and therefore needs to be validated through energy analysis (either MMPBSA/GBSA)

There should be one control docking score in the docking experiment. 

The overall manuscript is confusing and needs to be extensively revised.

Below are the detailed comments:

Abstract:

Abstract of the manuscript is vague and needs to be re-written. The methods used and results are summarized in the abstract, but the background needs to be properly introduced.

Line 14: Existing approaches for what? to understand the molecular mechanism of CSPs?

Line 15: olfactory interaction mechanisms: If the authors are trying to say interactions of olfactory machinery and odorants?

Introduction

line 36: you mean semiochemicals?

Line44: as per my knowledge ODEs are not transferred to the receptors rather they are used to degrade/remove the odorants from the vicinity of the ORs. I recommend authors to please correct this statement.

line 75: specific binding regions of what? CSPs??

Line 77 to 86: poorly written. Please re-write.  Write FPKM in full. To screen active molecules, protein models are used in this study, not the genes. What do you mean by PLIs are critical procedures, the receptor-ligand interactions give useful insights in understanding the molecular mechanism of the signaling process. The whole paragraph needs to be re-written by using appropriate terminologies.

Section 2.4:

Which template was used?

Section2.5:

I couldn’t get this protocol. How a production be done without minimization, heating and equilibrium. The system is first neutralized and then water model is added. Followed by energy minimization, heating, equilibrium, and production.

In methods, it is mentioned that trajectories are recorded every 2fs but in figure 8, ps is written, it is good to be consistent to avoid any confusion.

Section 3.1” what was the protein stability index? Are you sure protein was unstable?

Line 196: antenna recognition or odorant recognition within antenna?

Figure 4: the four cysteine should be highlighted in different colour.

Section 3.4: If there is any known ligand for this SnitCSP2, dock it to SnitCSP2 to have better idea of energy score as the energy scores with tested ligands are high. If no ligand data is available, I will recommend to use CSP2 from close homolog with experimental data to set as a control.

Section 3.5:

It is strongly recommended to execute MMPBSA or MMGBSA analysis to validate the CPS2 and ligand interactions.

The whole manuscript requires extensive English editing. Besides English, there are conceptual mistakes in the manuscript. 

Author Response

Dear Reviewer:

       Thank you very much for your kindly comments on our manuscript. There is no doubt that these comments are valuable and very helpful for revising and improving our manuscript. In what follows, we would like to answer the questions you mentioned and give detailed account of the changes made to the original manuscript.

    Regarding your overall evaluation of the article, we have made some adjustments and modifications after careful consideration. The simulation protocol has been modified and supplemented with specific changes in lines 180-197, and we have performed free energy calculations and per-residue contributions to the binding free energy of the SnitCSP2-ligand complex by the MM-PBSA method, with specific findings in lines 297-349. The modifications we made in the manuscript were highlighted in red.

Abstract of the manuscript is vague and needs to be re-written. The methods used and results are summarized in the abstract, but the background needs to be properly introduced.

Line 14: Existing approaches for what? to understand the molecular mechanism of CSPs?

1.Reply: Thank you for your significant reminder. Existing approaches refer to existing control measures that do not achieve good trapping effects, so it is expected to elucidate the molecular mechanism of protein-ligand binding for pest control. Change in lines 22-24.

Line 15: olfactory interaction mechanisms: If the authors are trying to say interactions of olfactory machinery and odorants?

2.Reply: Thank you for your significant reminder. Here it refers to the mechanism of action of external chemical stimuli and insect olfaction.

This section has been rewritten to make the summary clearer, with the main changes in lines 21-37.

Introduction:

line 36: you mean semiochemicals?

  1. Reply: Thank you for your reminder, this has been changed to “semiochemicals”.

Line44: as per my knowledge ODEs are not transferred to the receptors rather they are used to degrade/remove the odorants from the vicinity of the ORs. I recommend authors to please correct this statement.

  1. Reply: Thank you for your reminder, we have corrected this statement, the specific changes in lines 54-55.

line 75: specific binding regions of what? CSPs??

  1. Reply: This refers to the specific binding regions of SnitCSPs. the specific changes in line 86.

Line 77 to 86: poorly written. Please re-write. Write FPKM in full. To screen active molecules, protein models are used in this study, not the genes. What do you mean by PLIs are critical procedures, the receptor-ligand interactions give useful insights in understanding the molecular mechanism of the signaling process. The whole paragraph needs to be re-written by using appropriate terminologies.

  1. Reply: This section has been rewritten, with changes in lines 88-94.

Section 2.4:

Which template was used?

  1. Reply: The CSP6 crystal structure in Mamestra brassicae (PDB ID:1KX9, chain A) was used as a template. Already added here, change in lines 162-164.

Section2.5:

I couldn’t get this protocol. How a production be done without minimization, heating and equilibrium. The system is first neutralized and then water model is added. Followed by energy minimization, heating, equilibrium, and production.

In methods, it is mentioned that trajectories are recorded every 2fs but in figure 8, ps is written, it is good to be consistent to avoid any confusion.

  1. Reply: The protocol has been changed and added to, specifically in lines 180-197.

Section 3.1” what was the protein stability index? Are you sure protein was unstable?

  1. Reply: The probability of occurrence of certain dipeptides in stable and unstable proteins differs considerably, therefore, the frequency of occurrence of these dipeptides can be used to predict the stability of the protein in vitro. The smaller the instability coefficient, the more stable this protein is indicated. The instability index can be used as a reference value for the stability of a protein in in vitro tests. 40 or less suggests good stability, and greater than 40 suggests that the protein may be unstable. the specific changes in line 204.

Line 196: antenna recognition or odorant recognition within antenna?

  1. Reply: Thank you for your reminder. It refers to “odorant recognition within antenna”. the specific changes in line 218.

Section 3.4: If there is any known ligand for this SnitCSP2, dock it to SnitCSP2 to have better idea of energy score as the energy scores with tested ligands are high. If no ligand data is available, I will recommend to use CSP2 from close homolog with experimental data to set as a control.

  1. Reply: Sincerely thank you for your suggestion, based on your suggestion we did not find a reliable control template, and we are sorry that there is no way to implement this suggestion of yours. After reviewing the publications, we found that molecular docking experiments basically do not have a control group. We consider that the high or low values obtained from the docking of proteins with 14 ligands in the experiments are relative and that the sex pheromones, host plant volatiles and symbiotic fungal volatiles form a control with each other and are a comparative relationship, for example, the binding of SnitCSP2 to two female pheromones in the experiments is worse than other ligands. In general, for CSPs or OBPs docking experiments of species such as S.nitobei or its close relatives, S. noctilio, we generally consider that docking is good below -5.0 kJ/mol.

Section 3.5:

It is strongly recommended to execute MMPBSA or MMGBSA analysis to validate the CPS2 and ligand interactions.

  1. Reply: Sincerely thank you for your suggestion, we performed the free energy calculation and the contribution per residue to the binding free energy of the SnitCSP2-ligand complex by the MM-PBSA method, and the specific results are presented in 297-349.

Round 2

Reviewer 2 Report

Authors have addressed my earlier comments However, I am recommending few more minor changes to improve the manuscript. 

The first sentence of the summary is too long. Please split into two sentences.

Abstract first line: replace "resulting" with "causing"

Section 3.3: Mention exact sequence identity between target and template.

Figure 8: x-axis (residues should be till 120 but 12 is mentioned after 100)

Figure 10: It will be good to present ligand binding interaction diagrams for both pre-dynamics (post-docking) complexes as well as post-dynamics (after 40 ns production) and discuss if the binding residues have remained same after the dynamics or have changed.

Authors have improved the quality of English.

Author Response

Dear Reviewer:

     Thank you very much for your kindly comments on our manuscript. Regarding your overall evaluation of the article, we have made some adjustments and modifications after careful consideration. The modifications we made in the manuscript were highlighted in red.

The first sentence of the summary is too long. Please split into two sentences.

Reply: Thank you for your significant reminder, This sentence has been split into two sentences, with specific changes in lines 13-18.

Abstract first line: replace "resulting" with "causing"

Reply: Thank you for your reminder, Changes have been made here in the manuscript.

Section 3.3: Mention exact sequence identity between target and template.

Reply: The exact sequence identity has been added, specifically in lines 232-233.

Figure 8: x-axis (residues should be till 120 but 12 is mentioned after 100)

Reply: Thank you for the reminder that corrections have been made here.

Figure 10: It will be good to present ligand binding interaction diagrams for both pre-dynamics (post-docking) complexes as well as post-dynamics (after 40 ns production) and discuss if the binding residues have remained same after the dynamics or have changed.

Reply: Thanks to your very important suggestion, we have supplemented the revision with diagrams of ligand binding interactions for post-dynamics. The most stable conformation was selected for the visualization of ligand binding interactions by dynamics simulation, and this result is reliable. Compared with the interaction diagrams obtained for post-docking, some of the amino acids have changed, and the reason for the change is mainly because the dynamics is to simulate the trajectory of small molecules and obtain the whole process of docking between small molecules and proteins until stabilization, and all the conformations obtained are acquired at one instant in a continuous change, but the multiple conformations obtained by docking are discrete conformations, and there is no direct connection between any two, so the force diagram obtained is unreliable. However, due to the binding of the small molecule into the binding pocket of the protein, near many hydrophobic amino acid residues, some of the key amino acid residues remain unchanged. After consideration, the results of molecular docking are not reliable enough, so the manuscript does not present the ligand binding interaction diagrams after docking. This part is specifically changed in lines 339-356.